# Spring Wheat’s Ability to Utilize Nitrogen More Effectively Is Influenced by Root Phene Variation

**DOI:** 10.3390/plants12051010

**Published:** 2023-02-23

**Authors:** Rumesh Ranjan, Rajbir Yadav, Kiran B. Gaikwad, Naresh Kumar Bainsla, Manjeet Kumar, Prashanth Babu, Palaparthi Dharmateja

**Affiliations:** 1Division of Genetics, ICAR—Indian Agricultural Research Institute, New Delhi 110012, India; 2Department of Plant Breeding and Genetics, Punjab Agricultural University, Ludhiana 141004, India

**Keywords:** genetic variation, root traits, nitrogen use efficiency, wheat, high N, low N

## Abstract

Genetic improvement for nitrogen use efficiency (NUE) can play a very crucial role in sustainable agriculture. Root traits have hardly been explored in major wheat breeding programs, more so in spring germplasm, largely because of the difficulty in their scoring. A total of 175 advanced/improved Indian spring wheat genotypes were screened for root traits and nitrogen uptake and nitrogen utilization at varying nitrogen levels in hydroponic conditions to dissect the complex NUE trait into its component traits and to study the extent of variability that exists for those traits in Indian germplasm. Analysis of genetic variance showed a considerable amount of genetic variability for nitrogen uptake efficiency (NUpE), nitrogen utilization efficiency (NUtE), and most of the root and shoot traits. Improved spring wheat breeding lines were found to have very large variability for maximum root length (MRL) and root dry weights (RDW) with strong genetic advance. In contrast to high nitrogen (HN), a low nitrogen (LN) environment was more effective in differentiating wheat genotypes for NUE and its component traits. Shoot dry weight (SDW), RDW, MRL, and NUpE were found to have a strong association with NUE. Further study revealed the role of root surface area (RSA) and total root length (TRL) in RDW formation as well as in nitrogen uptake and therefore can be targeted for selection to further the genetic gain for grain yield under high input or sustainable agriculture under limited inputs.

## 1. Introduction

Wheat is consumed as a major dietary source of protein and carbohydrates by both humans and livestock [1] thus, the need to increase its production is undisputed. Globally, wheat occupies around 219 million ha of cropped area, which results in the production of around 760.9 million tons of wheat grains (FAOSTAT, 2020–2021). India is the second-largest producer of wheat grain (109.52 million tons, 2020–2021) after China, accounting for 14.3% of world wheat production. Increased nitrogen (N) fertilization besides dwarf and input responsive varieties were the major drivers of yield gain achieved during the second half of the 20th century [2] (Yadav et al., 2010). N is not only the most essential nutrient for higher grain yield realization but also a strong determinant for grain quality [3] (Marschner, 2012). Being a key element of protein, nucleotide, and chlorophyll, cereal crop plants use 20–50 g of N to produce a kilogram of biomass [4] (Robertson et al., 2009). N is the second most yield-limiting factor after soil water [5] (Campbell et al., 1993) and additional N applications are essential to optimize productivity and profitability. To carry out its essential functions, plants mostly absorb N from the soil and cannot use atmospheric N. Even among the available soil N, the plant can only absorb reactive N and soil does not carry enough reactive N (particularly nitrate and ammonia) to assist in plant development. Consequently, for crop production, farmers make use of water-soluble and readily absorbed chemical fertilizers. Global use of N fertilizer is continuously increasing and is estimated to be around 150 million tons/year by 2050 [6,7] (Good and Beatty, 2011; Ranjan and Yadav, 2019). Substantial leaching and loss of applied N (50–70 p%) from the plant–soil system due to volatilization, denitrification, and runoff have very large environmental implications such as an increase in greenhouse gas (N_2_O), algal bloom, and nitrate pollution in water bodies [8] (Malyan et al., 2016). It is difficult to assess the economic implications of inefficient N use, but around USD 2.3 billion can be redeemed for every one percent increase in NUE [9] (Raun and Johnson, 1999).

NUE is the total biomass production per unit of available N in the soil (Moll et al., 1982). NUE is the product of N uptake efficiency (NUpE) and N utilization efficiency (NUtE). Theoretically, NUE can be improved by increasing either NUtE or NUpE, or both. A NUE of around 33% has been estimated for cereal grain crops around the globe [9] (Raun and Johnson 1999). Exploration for plant genotypes with higher NUE, either physiologically (increased carbon (C) gain N^−1^) or agronomically (increased dry matter per unit N applied) is crucially important for sustainable agriculture [10] (Andrews et al., 2004).

In the conventional plant breeding approach, knowledge about the genetically exploitable variability, the genetic and physiological basis of variation in the working germplasm, is very important and according to [11] Foulkes et al., 2009, NUE can be improved genetically at least by 20%. The response of crop plants in terms of higher yield realization to an application of N is well-established [12] (Sinclair and Jamieson, 2006); it seems quite likely that selection for grain yield might have also impacted NUE directly or indirectly. Most of the earlier studies on NUE were either conducted on winter wheat comprising European and Chinese wheat germplasm or on the limited number of spring wheat, probably because of the lack of resources in developing world countries like India for such study and the difficulty in scoring root phenes. With significant genetic gain for wheat grain yield in the recent past along with no indication of yield saturation [13] (Yadav et al., 2021) targeting higher yield under high nutrient application with simultaneous buffering against weather vagaries is a common practice and target of wheat breeders and producers. Therefore, the present study was designed to study the variability available for root traits in advance wheat breeding germplasm *vis a vis* their role in nitrogen use efficiency through better uptake and or utilization. We hypothesize that the variability for traits such as NUE in our breeding materials, which have never been selected for or against the trait, should be sufficient for making a genetic gain.

## 2. Materials and Methods

### 2.1. Plant Materials

In this study, 175 spring wheat genotypes were used including advance breeding lines and commercially available Indian cultivars (Appendix A). All of the commercially released cultivars were developed under conventional tillage practices, whereas the advanced breeding lines were bred and selected under conservation agriculture environments.

### 2.2. Experiment I

#### 2.2.1. Hydroponics Plant Culture

During the winter season of 2014–2015, all 175 genotypes were raised in a hydroponic system in photoperiod and temperature-controlled environment at the National Phytotron Facility at IARI, New Delhi, India (Figure 1) under two different environments, namely, a high N (HN) and low N (LN) solution, with three replications. Healthy seeds of these genotypes were surface-sterilized for two minutes with one percent sodium hypochlorite before being rinsed with distilled water. A paper towel was used to germinate seeds in a seed incubator. Week-old seedlings were placed into plastic trays with a capacity of 18 L. Cotton plug-wrapped seedlings were placed on 8 mm diameter pre-drilled holes on acrylic leads of the plastic trays. The genotypes were grown at a temperature of 25 °C during the day and 22 °C at night, with a light intensity of 300 mol m^2^s^−1^ provided by cool fluorescent lamps in 10/14 h of dark and light timing, and relative humidity of 65–70%. The details of the nutrient solution, which included both N limiting and N non-limited environments are as described in [14,15] Ranjan et al., 19b, Ranjan et al., 2020. For the HN environment, the macronutrients were 0.4 mM NH_4_NO_3_, 10 mM KNO_3_, 2 mM CaNO_3_, 2 mM MgSO_4_, 0.1 mM KH_2_PO_4_, 1.5 mM CaCl_2_ and the micronutrients were 0.1 mM Fe-EDTA, 12.5 µM H_3_BO_3_, 2 µM MnCl_2_, 3 µM ZnSO_4_, 0.5 µM CuSO_4_, 0.1 µM Na_2_MoO_3_, 0.1 µM NiSO_4_, and 25 µM KCl was used as the nutrient medium. For the LN environment, N containing compound Viz., 0.4 mM NH_4_NO_3_, 10 mM KNO_3_, and 2 mM CaNO_3_ were reduced by one-quarter of HN. The solution was replaced every seven days to maintain the regular condition. For eight weeks, the seedlings were kept in hydroponic conditions.

#### 2.2.2. Measurements

The data on plant height (PH), maximum root length (MRL), root dry weight (RDW), shoot dry weight (SDW) and shoot nitrogen (N) percent were collected from8-week-old seedlings. Furthermore, the root–shoot ratio (R:S) was computed by dividing the root dry weight (RDW) by the shoot dry weight (SDW); and the total dry weight (TDW) was calculated by adding SDW and RDW. The NUE and its component traits (i.e., NUpE (nitrogen uptake efficiency) and NUtE (nitrogen utilization efficiency)) were calculated as per the procedure described in [14,16] Moll et al., 1982, Ranjan et al., 2019b.
$$\mathrm{Nitrogen}\;\mathrm{use}\;\mathrm{efficiency}\;(\mathrm{NUE}):\;\frac{\mathrm{Shoot}\;\mathrm{dry}\;\mathrm{weight}\;(\mathrm{gm})}{N\;\mathrm{supplied}\;\mathrm{in}\;\mathrm{gm}\;\mathrm{per}\;\mathrm{plant}\;}$$
$$\mathrm{Nitrogen}\;\mathrm{uptake}\;\mathrm{efficiency}\;(\mathrm{NUpE}):\;\frac{\mathrm{Total}\;N\;\mathrm{in}\;\mathrm{gm}\;\mathrm{per}\;\mathrm{plant}\;(\mathrm{gN})}{N\;\mathrm{supplied}\;\mathrm{in}\;\mathrm{gm}\;\mathrm{per}\;\mathrm{plant}\;}$$
$$\mathrm{Nitrogen}\;\mathrm{utilization}\;\mathrm{efficiency}\;(\mathrm{NUtE}):\frac{\mathrm{Shoot}\;\mathrm{dry}\;\mathrm{weight}\;(\mathrm{gm})}{\mathrm{Total}\;N\;\mathrm{in}\;\mathrm{gm}\;\mathrm{per}\;\mathrm{plant}\;}$$
where the total N in gm per plant (gN) = N % in shoot X× Shoot dry weight (gm). Thus, NUE=NUpE × NutE.

### 2.3. Experiment II

Based on the mean values for different traits, a selection of 19 genotypes (Appendix A) was made from 175 genotypes. This subset included four genotypes with the maximum N uptake, three genotypes with minimum uptake, the three best and three lowest-performing genotypes for N efficiency, three genotypes each with the maximum root weight, and the minimum root weight, respectively. To generate precise data on the NUE and related variables, these 19 genotypes were evaluated twice under hydroponic conditions as described in experiment I.

The plants were kept in hydroponic conditions for up to four weeks to collect comprehensive data on the root parameters in the selected genotypes. Roots of more than 4-week-old seedlings generally become intermingled in hydroponics systems, making it difficult to access the exact data on root traits. Except for the maximum root length (MRL), all other root traits such as total root length (TRL), root surface area (RSA), root volume (RV), average root diameter (AD), and the number of tips (N tips) were calculated by scanning the roots with a root scanner and a winRHIZO pro image analyzer (Regent Instruments Inc., Sainte Foy, QC, Canada) for each plant (Figure 2). MRL is the maximum length of root measured from the base of the stem to the longest tip of the root and was measured by a graduate ruler in centimeters (cm). The experiment was conducted twice, and the statistical analysis was based on the mean value of both experiments. Another hydroponic experiment with the same selection of genotypes was carried out for up to eight weeks to allow enough time for N uptake and utilization. After 4 days of drying in an oven at 60 °C, data were recorded on RDW and SDW. The Kjeldahl method was used to estimate the data on N% in the shoot. Other NUE components were determined using the NUE formula. To minimize the errors, the experiment was repeated twice, and the mean value of both experiments was used for further statistical analysis.

## 3. Statistical Analysis

The analysis of variance (ANOVA) along with interaction effects for 175 genotypes under HN and LN was carried out for the hydroponic experiment with GenStat release 14.1 software. Statistical software IndoStat version 9.2 was used to calculate various variability parameters viz., the phenotypic coefficient of variation (PCV), genotypic coefficient of variation (GCV), heritability (H^2^), and genetic advance (GA) as well as path analyses. The same program was used to conduct cluster analysis using Euclidean distance and average linkage. R program was used to determine the Karl Pearson correlation coefficient and principal component analysis. MS-Excel 2007 was used to create the graphical chart shown.

## 4. Results

Genetic variability for traits influencing NUE under hydroponic conditions.

### 4.1. Experiment I

#### 4.1.1. Analysis of Variance and Variability Parameters

The analysis of variance showed significant (*p* < 0.001) main effects due to genotypes in both high and low nitrogen environments for all of the traits studied (Table 1). The mean sum of the square of pooled data under both environments (i.e., under HN and LN) was significant (*p* < 0.001) for all of the traits studied. The variability parameter viz., mean performance, range, the genotypic coefficient of variation, phenotypic coefficient of variation, broad-sense heritability, genetic advance, and genetic advance percentage of the mean for various ten characters were analyzed and are presented in Table 2.

#### 4.1.2. Mean Performance, Broad Sense Heritability, and Genetic Advance

One of the primary component traits for NUE is SDW. In the HN environment, the average mean value for SDW was 0.617 gm, while in the LN environment, it was 0.333 gm (Figure 1). This trait’s range (Table 2) was 0.195 to 1.889 gm under HN and 0.08 to 0.78 gm under LN, indicating a 100% variance for this trait. This trait had a broad-sense heritability of 0.96 under HN and 0.98 under LN. Similarly, at 1% selection intensity, the genetic advance (GA) was 0.66 under HN and 0.33 under LN. Under HN, RDW had a mean value of 0.064 gm, whereas, under LN, it had a mean value of 0.066 gm. This trait had a GA of 0.095 in HN and 0.067 in LN. The R:S ratio is an important trait to consider when it comes to the portioning of assimilates in the root or shoot. Both under HN and LN, the H_bs_^2^ of the R:S ratio was found to be of the lowest value among the traits studied. All of the other component traits, particularly NUpE and NUtE, contributed to NUE. Under HN, the mean value of NUE was 26.8, while under LN, it was 58.4. Under HN, the range (Table 2) varied from 8.44 to 82.13, with the highest for the genotype HD2824/VL796 and the lowest for the genotype HD2967/DBW17. Similarly, under LN, the range varied from 14.04 to 136.0 with a maximum for the genotypeHD2967/HD3024 and a minimum for the genotype PBW 621-50. In comparison to HN (28.8), the GA appeared to be more under LN (58.8) for NUE.

#### 4.1.3. Diversity Analysis of Genotypes under High N

A total of 175 genotypes of spring wheat evaluated under the study were classified into VI clusters under HN and are presented in Appendix A. The majority of the genotype grouped into a single cluster. Cluster I had a maximum (160 genotypes) number of genotypes followed by Cluster IV (11 genotypes). Clusters II, III, V, and VI accommodated one genotype each, respectively. Appendix A represents the average intra and inter-cluster D^2^ values for all of the VI clusters. These cluster values were calculated according to the method given by [17] Singh and Chaudhary (1977).

#### 4.1.4. Diversity Analysis of Genotypes under Low N

The 175 spring wheat genotypes were classified into ten clusters under LN, as illustrated in Appendix A. Cluster I has the most number of genotypes (136), Cluster II has 22 genotypes, and Cluster III has ten genotypes. The rest of the genotypes were accommodated in Clusters IV, V, VI, VII, VIII, IX, and X, respectively. The average intra and inter-cluster D^2^ values are shown in Appendix A for all ten clusters. A subset of six genotypes was chosen based on their NUE, with the three best (from cluster IX) and three lowest-performing genotypes (cluster I).

#### 4.1.5. Principal Component Analysis

In this study, ten variables were classified into three and four principal components under HN and LN, respectively, explaining 94 %and 93 %of variation (Figure 2). Only the top three PCs were considered, which explained 86%(HN) and 82%(LN) of the total variance, respectively. SDW, RDW, gN, NUpE, and NUE accounted for the maximum variation in HN and were grouped in the first principal component (Appendix A); however, the same variables also accounted for the maximum variation in LN (Appendix A). Under HN, SDW, NUE, NUpE, and gN were highly correlated with PC1, whereas RDW, RL, and R:S were significantly correlated with PC2, according to the correlation coefficient between the main components and variables (Figure 3). Similarly, under LN, SDW, RDW, gN, NUpE, and NUE were highly correlated with PC1, but NUtE was highly correlated with PC2.

#### 4.1.6. Association Analysis

The correlation coefficient was determined between the various contributing traits for NUE under HN and LN (Figure 4). In the current study, NUE had a substantial positive correlation with SDW (0.996), NUpE (0.936), RDW (0.504), plant height (0.385), and MRL (0.264) in an HN environment, but R:S had a negative significant correlation (−0.244). There was no substantial association between the shoot N percent to NUtE and NUE. Similarly, a correlation among various NUE traits could be deciphered from Figure 4. Under the LN environment, NUE showed a positive and significant association with RDW (0.979), NUpE (0.868), MRL (0.356), and plant height (0.294), and no correlation was found with N% in the shoot, R:S, and NUtE, respectively.

When compared to all other traits, RDW had the strongest and most significant correlation with NUE. It seems to be possible that additional components of root traits, in addition to RDW, contribute to high RDW. As a result, we chose to examine the other root traits and generated data on a subset of 19 genotypes.

### 4.2. Experiment II

#### 4.2.1. Root Traits of the Subset Genotypes under HN and LN Environments

##### Mean Performance

The experiment was conducted twice, and the results are presented as a pooled mean and range for different root traits (Table 3).The overall mean of the genotypes for RSA was 41.45 cm^2^ under HN and 62.91 cm^2^ under LN (Figure 5). This trait range was 20 to 64 cm^2^ under HN conditions and 30 cm^2^ to 153 cm^2^ under LN conditions. Under HN, the released variety HD 3090 (64 cm^2^) had the highest RSA and PBW 621-50 (20 cm^2^) had the lowest, whereas, under LN, HD2824/VL796 (153 cm^2^) had the highest and PBW343/PICCI LOCAL/RL6080 had the lowest (30 cm^2^). Furthermore, as shown in Figure 5, the mean data for MRL suggests a larger value under LN (27 cm) than HN (21 cm), which was concurrent to a mean value of TRL expressed more under LN (515 cm) than HN (425 cm). Under LN and HN conditions, the RDW mean data revealed no differences. Under HN, the value ranged from 0.01 gm (HD2967/HD3035) to 0.033 gm (HD2967/DT2761), whereas under LN, the value ranged from 0.005 gm (HD2967/HD3024) to 0.023 gm (DL672/P66.270/DE894/3/CUMMYN).

##### Association Analysis

Correlation analysis was used to determine the relationship between and among different root and shoot traits. RDW exhibited a positive and significant correlation with RSA (0.603), RV (0.519), TRL (0.741), and SDW (0.637) in subset genotypes under HN (Figure 6). Though statistically non-significant, a higher correlation coefficient (0.443) implied a broad relationship between RDW and N tips. RDW, on the other hand, had no association with AD or MRL. SDW showed a positive and significant correlation with RSA (0.569), RV (0.620), and TRL (0.669). SDW had a non-significant association with other features. Traits such as RSA, RV, N tips, and MRL showed a positive correlation with each other, whereas MRL showed no correlation with all the other traits.

Under LN, RDW had a positive and significant correlation with RSA (0.526), RV (0.480), AD (0.457), TRL (0.549), and SDW (0.514) (Figure 6). Similarly, RSA (0.472), RV (0.476), AD (0.444), N tips (0.327), and TRL (0.465) all had a positive correlation with SDW. Similar to HN, RSA, AD, RV, and TRL also had a positive correlation with each other. All other traits hada non-significant correlation with MRL.

##### Path Analysis

The direct and indirect effects of various traits on RDW under HN and LN were analyzed and presented in Table 4 and Table 5.

Under HN, TRL (1.722), followed by RSA (0.642), had the maximum direct effect on RDW. Other component traits contributed largely through an indirect route. RV made a significant contribution to RDW mostly through RSA and TRL. Similarly, under LN, TRL (0.515) and SDW (0.472) both contributed directly to RDW. Other factors had a negligible positive relationship with RDW.

#### 4.2.2. Validation of Traits Contrivance for NUE in a Subset of Genotypes Was Grown for Eight Weeks

The correlation coefficient analysis in the subset genotypes (Figure 7) showed that NUE had a significant positive association with RDW (0.769), MRL (0.588), and NUpE (0.961) under the HN condition. The relationship between NUtE and NUE was inverse. NUpE (0.793), MRL (0.823), and R:S (0.841) all had a positive association with RDW, whereas NUpE had a negative relationship with NUtE. (−0.231). Under the LN condition, NUE also had a significant positive relationship with NUpE (0.967) and RDW (0.745).The objective of this study was to confirm the findings arising from the subset genotypes for NUE traits to the findings of experiment I, which included 175 genotypes. The data demonstrated that, when compared to the other component traits, the RDW had the strongest and most significant association with NUE in both HN and LN.

To find the correlation between the hydroponic and soil data, an experiment with a PVC pipe filled with soil was conducted (Figure 3). The results showed a good accordance between the hydroponic and pipe filled with soil data for the root and NUE traits, which was published [14] (Ranjan et al., 2019b).

## 5. Discussion

The use of nitrogenous fertilizer has increased at a much faster rate than grain yield in most cereal crops. The increasing concern about environmental pollution caused by unused N fertilizer has forced global policy planners to incentivize the development of nitrogen use efficient varieties. The existence of genetic variability for NUE and its subsequent exploitation, even in major crops such as wheat, for the development of N-efficient varieties has been largely ignored because of the cheaper cost and abundant supply of N sources. NUE is a complex trait dependent upon a number of component traits such as nitrogen-capturing root traits, optimum portioning of captured N in different organs, and efficient utilization for higher yield realization. Thus, genetic improvement for NUE requires thorough study and an understanding of the various contributing trait.

The analysis of variance showed significant differences among the genotypes, and environments with a strong genotype X environment interaction for all of the traits under study. Since the material under study comprises the advance breeding lines of a breeding program mainly focused on yield enhancement and better adaptation, along with some released varieties with no history of direct selection for N capturing root traits or other physiological processes relevant for N utilization, an abundance of variability for various traits related to NUE in the material is natural. However, the indirect effect of selection for better physiology for grain yield cannot be ruled out as N is an important element involved in every path relevant for yield formation. A strong genotype X environment (high and low N) interaction suggests a separate breeding program for each environment. The existence of sufficient genetic variation for traits relevant to NUEis also reported in several other cereal crops [18,19,20,21,22,23,24,25] (Ortiz-Monasterio et al., 1997; Foulkes et al., 1998; Rakotoson et al., 2017; Nehe et al., 2018; Almeida et al., 2018; Ranjan et al., 2019a; Mălinas et al., 2022, and Decouard et al., 2022). Breeders are always in search of an environment that helps in differentiating their breeding material for the traits under selection, and the present study indicates that LN was more effective in differentiating the genotypes for root traits relevant for N capturing. Selection under low N in a large number of early fixed generations for root traits and subsequent verification of only selected lines under HN can be highly rewarding for resource-limited breeding programs.

The large range for the majority of traits under study provides enough scope for genetic manipulation. The effectiveness of selection for difficult-to-score root traits and a derivative trait such as NUE can be improved through hydroponic screening as it minimizes the environmental impact, strongly indicated by the narrow difference between the GCV and PCV values in the presented study. Our results corroborate earlier findings [26,27,28,29] (Gaju et al., 2011; Petrarulo et al., 2014; Ahmadi et al., 2018 and Yin et al., 2018). Even the genetic variation expression for different traits also varied over the environment, for example, the R:S ratio was better expressed under high HN whereas NUpE was better expressed under LN. However, under both environments, variation expression for RDW was sufficiently large and can be effectively targeted for selection under both conditions. Direct selection in early fixed generation (F_5_ and F_6_) for most of the root traits with sufficiently high heritability under hydroponics can be effectively integrated with most wheat breeding programs. High heritability for various traits relevant to NUE has also earlier been reported in maize by [30] Presterl et al., (2003) and in wheat by [31] Laperche et al., (2006). To improve selection efficiency, the selection of component traits such as SDW, MRL, RDW, and R:S has also been emphasized by [32] Sathish et al., 2016 and [33] Naveen and Uma, 2016. In the absence of suitable screening techniques for root phenes [34] (Whalley et al., 2017), root traits have rarely been targeted in breeding programs. Scoring in hydroponic conditions for root biomass and root architecture, at least in the early advanced bulk, can facilitate the large-scale selection of both N responsive and N efficient genotypes, as indicated by the significant correlation between the hydroponic and pipe-filled with soil data for NUE traits [14] (Ranjan et al., 2019b).

Despite the large variability indicated by ANOVA and the magnitude of D^2^ values, clustering based on root phenes and other component traits for NUE grouped the majority of the genotypes in a single group in both environments. The selection of parents within a cluster or from different clusters depends upon the breeding objectives and the genetic basis of variability. Germplasm line 154 (HD2824/VL796) (Appendix A) accommodated in cluster V had favorable phenological traits probably because of the right type of *vrn* alleles. The role of the *vrn* gene in balancing the above- and below-ground plant architecture is well-documented [35] (Deng et al., 2015)). The germplasm evaluated in the study is an outcome of a single breeding program with a major focus on yield gain, so this type of group is not surprising. However, segregants for desirable root phenes can be created by crossing the lines within a group or by a line complementing root phenes with line number 154 under HN. The low nitrogen condition discriminates the genotypes under testing more effectively, providing stronger help in the selection of parents for character complementation for desirable root phenes. Germplasm line 86(HD2967/HD3024), with a majority of the traits relevant for nitrogen capturing and utilization, can be effectively crossed with the desirable parents from any other clusters. Though the role of the size of the root system on nutrient mining is invariably agreed upon, there is still no consensus on the root architecture for better NUE [36] (Palta, 2011). It was, therefore, very pertinent for us to identify the optimum root phenes for NUE [37] (Hawkesford, 2014).

In the present investigation, we found a significant positive correlation of NUE with NUpE, RDW, PH, and MRL under both HN and LN environments. Furthermore, the association between NUE and NUpE was much stronger under the HN than LN conditions. Time series analysis of released varieties in Mexico [18] (Ortiz-Monasterio et al., 1997) indicated that genetic gains in NUE under low N were largely because of the improvement in NUpE in contrast to NUtE in France [38] (Brancourt-Hulmel et al., 2003) and the UK [19] (Foulkes et al., 1998). Contrasting results were also indicated under HN conditions with equal importance of NUpE and NUtE in Mexico [18] (Ortiz-Monasterio et al., 1997) and Finland [39] (Muurinen et al., 2007), while there was more importance of NUpE than NUtE in Australia [40] (Sadras and Lawson, 2013). Broadly, these reports indicate that the genetic variation in NUE was more commonly associated with NUpE at low N supply, while it was more commonly associated with NUtE at high N supply. In our study, NUpE was found to be more relevant under both the HN and LN conditions because of the comparatively smaller time available in the spring wheat genotypes for efficient exploration of the soil profile during the vegetative phase in contrast to winter genotypes. Any genotype, therefore, with large root biomass, was able to explore a larger soil volume in a limited time. In spring wheat, the genotype with a mild vernalization requirement has a comparatively longer vegetative period, particularly under early seeding, and therefore keeps on accumulating more root biomass and thereby higher NUE. Our results corroborate the earlier findings in wheat [21] (Nehe et al., 2018); maize [41,42] (Bertin and Gallais, 2000; Gallais and Hirel, 2004) and in rice [20] (Rakotoson et al., 2017). A smaller role of NUtE in NUE in both the low and high N conditions may be due to reduced assimilation of ammonium in amino acids. NUtE is more related to enzymes involved in the reduction and incorporation of nitrogen in organic compounds such as nitrate reductase (NR), nitrite reductase (NiR), and glutamine synthetase (GS) [7] (Ranjan and Yadav, 2019). The exact reason for the poor association of NUtE’s poor association with NUE can therefore be explained through further studies into the genetic variation of these key enzymes and their interactions with primary growth regulatory genes such as *Vrn1*. Studies in rice [39] (Muurinen et al., 2007) and barley [43] (Sinebo et al., 2004)) have also shown that NUpE is more important than NUtE in determining NUE.

Larger and deep-rooted systems generally have better foraging capabilities, even from the deeper soil profile, and thereby the root: shoot ratio has always been considered a criterion for improving drought and nutrient use efficiency [44] (Srividhya et al., 2011). NUpE is greatly influenced by the root phenes and their biomass, especially under N limitation, as plants direct their assimilate toward the production of root biomass for better exploration [45] (Hermans et al., 2006). With more extraction of nutrients through a better root system, the plant can produce more shoot biomass, and it was because of this reason that we found a positive association between the root and shoot biomass. The negative association between NUpE and NUtE under both LN and HN corroborates the earlier finding of [42] Gallais and Hirel, (2004) in maize and [46] –Hitz (2015) in wheat. This can either be due to a variable proportion of N transporter present in the root hairs and enzyme involved in N assimilation or might be due to the degradation of leaf protein (Rubisco). Moreover, the poor role of NUtE for NUE in the present experiment might also be due to the early termination of the experiment.

As root biomass was found to play an important role in N foraging and uptake, the important question before us was to find the root traits responsible for higher root biomass. Since these root traits are very difficult to score, even under hydroponic conditions, we selected a subset of genotypes based on their root biomass. Moreover, these traits are highly influenced by variable environments imposed by heterogeneous soil [47] (Lynch, 2007). Root image analysis of the sample generated through hydroponics can encourage the breeder to integrate these into their breeding program. Therefore, with low cost and no confounding by the environment, hydroponic screening can be very effective in quantifying the genetic basis for differences in root features [14] (Ranjan et al., 2019b). Path analysis revealed a strong direct effect of TRL and RSA on RDW, therefore suggesting that more emphasis on these two traits during selection in an early segregating generation can provide a stronger genetic gain. Our research revealed that direct selection for RDW to enhance NUE in wheat under Indian conditions is rather simple and can also be extremely rewarding in improving NUpE. If the resources available to the breeding group are good, then integration of the selection for TRL, RV, and RSA in the early fixed material can improve the genetic gain for NUE. Furthermore, [48] Caassen and Barber (1976) and [49] Imada et al., (2008) also indicated the importance of RSA. Higher root weight results in healthier above-ground biomass due to improved absorption and C fixation, and vice versa [50] (McPhee, 2005). Better root exploration and nutrient uptake boost the photosynthetic process, resulting in more assimilation, which is again shared by the roots and shoots. TRL and RDW, according to [51] Kumar et al., (2012), produced the most phenotypic diversity in the root system and may be adequate to improve other root attributes. The direct selection of root traits including the length, area, and volume not only benefits the N uptake, but also the water absorption in wheat [52] (Hurd, 1964), upland rice [53] (Price et al., 2002), and maize [47] (Lynch, 2007).

## 6. Conclusions

The germplasm under study shows the existence of sufficient genetic variability for the shoot and root traits relevant to NUE.A low nitrogen environment is more effective in discriminating the genotypes for root traits compared to a HN environment.RDW was found to be highly associated with better NUE under both nitrogen-rich and nitrogen-poor environments.Under N poor environment, the maximum root length can be directly selected to improve the NUE whereas N uptake can be improved by directly selecting the root biomass under an N-rich environment.Selection for root biomass is not in conflict with above-ground biomass, which is highly relevant for higher yield realization in the future.

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
