# Peer review of "Spring Wheat’s Ability to Utilize Nitrogen More Effectively Is Influenced by Root Phene Variation"

_plants, 2023, doi:10.3390/plants12051010_

Round 1
Reviewer 1 Report
Overall, the manuscript is written well however few changes can improve the manuscript.
It is interesting to see that evaluation was done in both hydroponics ( controlled environment) and in the field.
Title: Can be better, suggest a different word for Contrivance
Line item 18, 35, 41 and many places: Check spacing and proof reading will improve the quality of this manuscript.
Line item 99: How did you come up with high nitrogen and low-nitrogen treatments for hydroponics? How are your treatments are relevant to field low and high nitrogen? I did not notice the design (Randomized complete block design?).
Line item 132: Can you elaborate maximum root length?
Line item 135: You had used winRHIZO to scan the roots. I believe root system would be extensive for 8 week old plants. How did you overcome this while scanning roots and getting all the root data?
Conclusion: It can be improved further.
Thank you.
Author Response
Dear Reviewer,
Thanks for your comments and suggestions. The response to the comments has been tried to satiate as below. The revision of the manuscript has been done in track change in Ms word. I hope the response to your queries meets your requirement. I would like to thank you again for your queries to improve the quality of the manuscript. Thanks
Response to comments
Reviewer 1
Title: Can be better, suggest a different word for Contrivance
Reply Thanks for the suggestion. The suggested title may be as: “Root phene for improving nitrogen use efficiency in spring wheat”
Line item 18, 35, 41 and many places: Check spacing and proof reading will improve the quality of this manuscript.
Reply Thanks for the suggestion. Checked and corrected as suggested.
Line item 99: How did you come up with high nitrogen and low-nitrogen treatments for hydroponics? How are your treatments are relevant to field low and high nitrogen? I did not notice the design (Randomized complete block design?).
Reply Thanks for the comment. The protocol for nutrient medium under high and low N has been mentioned in manuscript lines No. 98-104. The protocol for the nutrient medium was followed by Sharukov et al. 2012 which reflects the same as of field condition. For HN, we have doubled the concentration and for LN, we reduce the concentration to half. Since the experiment was performed under artificially created hydroponic conditions in National Phytotron Facility to minimize the environmental error due to soil heterogeneity of the field and get proper root traits data of the genotypes under evaluation. However, we have randomized the treatments in the second replication that follows CRD.
Line item 132: Can you elaborate maximum root length?
Reply Thanks for the comment. Elaborated as suggested in line no. 133-134.
Line item 135: You had used winRHIZO to scan the roots. I believe root system would be extensive for 8 week old plants. How did you overcome this while scanning roots and getting all the root data?
Reply Thanks for the query. The root systems of 4 weeks old plants under hydroponic were compatible with our scanning area of winRHIZO appliance and were able to get all the root traits data important for our studies without difficulties whereas, for 8-week-old plants, we have only recorded MRL and root biomass due to difficulties in scoring other root traits (mentioned in line 134-135). No root scan data were recorded for 8 weeks old plants.
Conclusion: It can be improved further.
Reply Thanks for the comment. The conclusion has been modified to this extent.

Reviewer 2 Report
This manuscript title “Contrivance of root traits in adaptation to nitrogen use efficiency in spring wheat” I suggest that the "Results" section should be reorganized and rewritten and the manuscript requires rewriting and polish by a professional English editor.
1. “Introduction” please use the correct reference format.
Line 64:“ N-1 ” –it should be “N-1”
Line 69, 78:more spaces
I suggest that the scientific hypothesis should be rewritten more appropriately
2. The "Materials and Methods" section is not clear. I suggest that the author should describe the material and methods section in more detailed. It is hard to follow the method since some essential information is missing.
It is not cleat at all what was the reason of having a hydroponics experiment on following with pot experiment?
Pot size under hydroponics experiment and pot experiment?
Line 119-120, 141:the calculation method? please explain in a more descriptive way
Line 103, 107:more spaces
Line 105:“ m 2 s-1” –it should be just “m 2 s-1”
Line 119-120:please explain in a more descriptive way
Line 103, 107:more spaces
Line 151:why only two repetitions?
Experiment3:Sample harvest method?
3. The "Results" part needs to be significantly improved, it is confused and needs to be reorganized and rewritten.
Figure: the X-axis and Y-axis are missing
Line 171,173: “(p 0.001)” –it should be “(p <0.001)”
Line 209-210: different font sizes
Line 232-234: it should be in the introduction.
Data source for each graph? Experiment 1? Experiment 2? Experiment 3?
What are the results of Experiment 3?
Each illustration should be explained in detail with legend, and should be autonomous and understandable without a search of explanations in the text.
4:The “Discussion” part is insufficient, this section is written not as a discussion but as an introduction.
Author Response
Dear Reviewer,
Thanks for your comments and suggestions. The response to the comments has been tried to satiate as below. The revision of the manuscript has been done in track change in Ms word. I hope the response to your queries meets your expectation. I would like to thank you once again for your comments/queries to improve the quality of the manuscript. Thanks
Response to comments
Reviewer 2
- “Introduction”
please use the correct reference format.
Reply Thanks for the suggestion. The reference format in the manuscript follows the same pattern.
Line 64:“ N-1 ” –it should be “N-1”
Reply Thanks for the suggestion. Corrected as suggested.
Line 69, 78:more spaces
Reply Thanks for the suggestion. Proper spacing has been inserted/corrected in the text where ever required.
I suggest that the scientific hypothesis should be rewritten more appropriately
Reply Thanks for the comment. The Scientific hypothesis has been rewritten in line no. 78-80.
- The "Materials and Methods" section
It is not clear at all what was the reason of having a hydroponics experiment on following with pot experiment?
Reply Thanks for the query. The hydroponic experiment was performed to minimize the environmental error due to soil heterogeneity in the field for the NUE experiment as the hydroponic experiment was conducted under an artificially created environment at National Phytotron Facility.
- A pot experiment was performed to find the corollary between hydroponic and soil data.
Pot size under hydroponics experiment and pot experiment?
Reply Thanks for the comment. The pot size of PVC pipe is mentioned from line no. 146-147.
Line 119-120, 141:the calculation method? please explain in a more descriptive way
Reply Thanks for the comment. The calculation method has mentioned in the manuscript line no. 113-117.
Line 105:“ m 2 s-1” –it should be just “m 2 s-1”
Reply Thanks for the suggestion. The correction has been done in the manuscript as suggested.
Line 151:why only two repetitions?
Reply Thanks for the query. To establish the corollary between the root data of hydroponic and soil, we have planned an experiment to plant the subset genotypes in PVC pipe. Due to high soil heterogeneity and lateral leaching of N from one field to another nearby field, it seems difficult to maintain the low N environment in field conditions,s and at the same time, studying the root traits was highly labor oriented in the field. Therefore, to minimize the experimental error to an extent, we have opted for two replication under HN and LN.
Experiment3:Sample harvest method?
Reply Thanks for the query. The result of experiment 3 has been published earlier by Ranjan et al., 2019b entitled Ranjan, R., Yadav, R., Kumar, A. and Mandal, S.N. (2019b). Contributing traits for nitrogen use efficiency in selected wheat genotypes and corollary between screening methodologies. Acta Agriculturae Scandinavica, Section B—Soil & Plant Science, 69(7), 588-595.
- The "Results" part needs to be significantly improved,
Line 171,173: “(p 0.001)” –it should be “(p <0.001)”
Reply Thanks for the suggestion. Corrected as suggested.
Line 209-210: different font sizes
Reply Thanks for the suggestion. Corrected as suggested.
Line 209-210: different font sizes
Reply Thanks for the suggestion. Corrected as suggested.
Line 232-234: it should be in the introduction.
Reply: Thanks for the suggestion. The sentence has been omitted.
The data source for each graph? Experiment 1? Experiment 2? Experiment 3?
Reply Thanks for the comment. The graph of Experiment 1, 2 and 3 have been illustrated from the recorded data.
What are the results of Experiment 3?
Reply Thanks for the query. The result of experiment 3 has been published earlier by Ranjan et al. 2019b. The part of the finding has been mentioned in the manuscript i.e. line no. 270-272.
Each illustration should be explained in detail with legend, and should be autonomous and understandable without a search of explanations in the text.
Reply Thanks for the comment. The legend has been added where required in the figure.
4:The “Discussion” part is insufficient, this section is written not as a discussion but as an introduction.
Reply Thanks for the suggestion. The discussion part has been rewritten and modified to an extent.

Round 2
Reviewer 2 Report
The manuscript was improved satisfactory. However, some issues are still of concern:
A general comment: The manuscript is an unfinished revision and it is difficult to evaluate revisions in the text.
1. Line 119-123:Equations should be inserted in editable format from the equation editor.
2. In comment:“It is not cleat at all what was the reason of having a hydroponics experiment on following with pot experiment?”the authors explained clearly the reason. However, this reason is not added to the context. It is suggested to add and the readers could better understand.
3. The result of experiment 3 has been published earlier. I suggest that the authors delete experiment 3 on the material and method section.
4. Fig 1 and Fig 5: lack of statistical analysis, which can add the significant letters.
Author Response
Dear Reviewer,
Thanks for your comments and suggestions. Those suggestions help to improve the quality of the manuscript and information generated from the experiments can be transferred to the reader in an understandable manner. Some of the queries has been arose and tried to improve accordingly. I would like to thank you once again for your untiring effort. Thanks
Comments and Suggestions for Authors
- Line 119-123:Equations should be inserted in editable format from the equation editor.
Reply: Thanks for the suggestion. The equation editor seems not compatible with the downloaded manuscript for revision. I have tried to have an equation editor, but it seems not consistent with the current revised manuscript word document. I'll keep it in mind and make it in the suggested format once the galley proof is finalized by the publisher.
- In comment:“It is not cleat at all what was the reason of having a hydroponics experiment on following with pot experiment?”the authors explained clearly the reason. However, this reason is not added to the context. It is suggested to add and the readers could better understand.
Reply: Thanks for the suggestion. The suggestion has been added in manuscript line no. 273 – 275.
- The result of experiment 3 has been published earlier.I suggest that the authors delete experiment 3 on the material and method section.
Reply: Thanks for the suggestion. As suggested, experiment 3 is removed from MM section.
- Fig 1 and Fig 5: lack of statistical analysis, which can add the significant letters.
Reply: Thanks for the suggestion. Figure 1 and Figure 5 are the diagrammatical representation of mean value ± SE of respective traits under High and Low Nitrogen environments.